# Emission characteristics of greenhouse gases and air pollutants in a Qinghai-Tibetan Plateau city using a portable Fourier transform spectrometer and TROPOMI observations

Qiansi Tu[1], Frank Hase[2], Ying Zhang[3], Jiaxin Fang[1], Yanwu Jiang[1], Xiaofan Li[3], Matthias Schneider[2], Zhuolin Yang[3], Xin Zhang[4], Zhengqiang Li[3]

[1]School of Mechanical Engineering, Tongji University, Shanghai 201804, China

[2]Institute of Meteorology and Climate Research (IMK-ASF), Karlsruhe Institute of Technology (KIT), Eggenstein-Leopoldshafen 76344, Germany

[3]State Environmental Protection Key Laboratory of Satellite Remote Sensing, Aerospace Information Research Institute, Chinese Academy of Sciences, Beijing 100101, China

[4]College of Resources and Environment, University of Chinese Academy of Sciences, Beijing 101408, China

*Correspondence to*: Zhengqiang Li (lizq@radi.ac.cn)

## Abstract

Despite the critical need to understand greenhouse gas and air pollutant concentrations and their emissions characteristics in urban and industrial areas, limited assessments have been conducted in the Qinghai-Tibetan Plateau (QTP) cities. Herein, for the first time, we present $CO_2$, $CH_4$ and CO column abundances using a portal Fourier-transform infrared spectrometer (EM27/SUN) in Ganhe Industrial Park (36.546°N, 101.518°E, 2603 m a.s.l.), located in the suburbs of Xining, Qinghai Province, during May – June 2024. Ground-based measurements found to be higher than spaceborne measurements (TROPOMI and IASI) and model forecast (CAMS) across all investigated species, indicating higher local emissions. Notably, significant discrepancies in CO levels are observed, particularly under easterly wind conditions, which transport polluted airmasses from Xining city. To further quantify emissions, we applied a simple dispersion model to the EM27/SUN data and TROPOMI products, estimating an average CO emission rate of $12.3 \pm 9.6$ kg/s and $8.9 \pm 7.5$ kg/s, respectively. A wind-assigned anomaly method further applied to the TROPOMI dataset yielded a CO emission rate of 8.5 kg/s. Additionally, the ground-based observations of $\Delta XCO/\Delta XCO_2$ ratio exhibits a strong correlation under easterly winds, which suggests an average $CO_2$ emission rate of 550 kg/s from Xining city. These findings underscore the utility of portable FTIR spectrometers to enhance our understanding of urban emissions at QTP and demonstrate the potential of combining collaborative ground-based and spaced-based observations to estimate $CO_2$ emissions, particularly in regions with sparse $CO_2$ measurement coverage.

## 1. Introduction

Carbon dioxide ($CO_2$) and methane ($CH_4$) are two primary greenhouse gases (GHGs) whose atmospheric concentrations have surged to unprecedented levels since 1750, largely driven by anthropogenic activities. According to the WMO's annual Greenhouse Gas Bulletin (https://library.wmo.int/idurl/4/69057, last access: January, 2024), the globally averaged surface concentration of $CO_2$ reached 420 parts per million (ppm), and $CH_4$ reached 1934 parts per billion (ppb) in 2023. Carbon monoxide (CO) is one of the most important atmospheric pollutants, which is primarily produced by inefficient combustion, such as biomass burning (Griffin et al., 2024), traffic and industrial activity (Dils

et al., 2011). CO plays a crucial role in atmospheric chemistry, especially in the troposphere, where it reacts with the hydroxyl radical (OH) in the reaction $CO + \cdot OH \rightarrow CO_2 + \cdot H$ (Spivakovsky et al., 2000; Thompson, 1992). This oxidation process serves as the dominant sink for atmospheric CO, accounting for 90-95% of its total removal. Consequently, CO indirectly affects the lifetime of $CH_4$ by reducing the availability of OH that would otherwise oxidize $CH_4$. This, in turn, indirectly contributes to global warming (IPCC, 2007).

To achieve highly accurate and precise measurements of total column abundances of $CO_2$, $CH_4$ and CO, the Total Carbon Column Observing Network (TCCON) and the Collaborative Carbon Column Observing Network (COCCON) have been established. Both networks utilize Fourier-transform infrared (FTIR) spectrometers. TCCON employs high-resolution FTIR instruments for exceptionally precise $CO_2$, $CH_4$ and CO measurements amongst other gases (Wunch et al., 2011). In contrast, COCCON employs portable, low-resolution FTIR spectrometers to measure $CO_2$, $CH_4$ and CO, providing a valuable complementary extension to the TCCON network (Alberti et al., 2022; Frey et al., 2015; Herkommer et al., 2024).

The Qinghai-Tibetan Plateau (QTP), with its unique topography, plays a crucial role in Earth's climate system and has become a key region for monitoring climate trends and global air quality (Kang et al., 2021; Zhang et al., 2022). A recent study highlights that surface pollutants in Asia can be transported into the stratosphere from the Tibetan Plateau region, potentially exerting significant impacts on global climate (Bian et al., 2020). There are only two stations on the Tibetan Plateau—Waliguan, the only global station in Eurasia, and Shangri-La—that measure surface concentrations of $CO_2$, $CH_4$, and CO, making them important for understanding atmospheric composition in this critical region (Guo et al., 2020; Xiong et al., 2022). These stations conduct in-situ measurements and provide highly accurate surface observations and valuable insights into local fluxes; however, they are influenced by surface exchanges and vertical transport, which can limit their ability to estimate sources and sinks over larger spatial scales. When combined with other observation types, such as FTIR, which capture emissions and transport on a broader scale, they become complementary, together offering a more comprehensive understanding of sources and sinks at both local and regional levels (Callewaert et al., 2022). Additionally, surface observations are less effective for satellite validation compared to column-based measurements, such as those obtained from TCCON and COCCON. However, there are currently no TCCON or COCCON stations on the QTP. Zhou et al. (2023) conducted the first FTIR column observations in a small city on QTP, but measurements in larger cities, such as Xining—the capital of Qinghai Province, remain sparse. This is notable since most anthropogenic emissions of GHGs and CO are concentrated in urban centers (Crippa et al., 2021).

This critical observational gap extends to space-borne measurements over the topographically complex QTP, which could provide the broad spatial coverage lacking from ground-based networks. However, current satellite capabilities reveal complementary limitations: while the TROPOspheric Monitoring Instrument (TROPOMI) offers extensive spatial coverage for CO measurements, it does not provide $CO_2$ data. Conversely, satellites specializing in greenhouse gases like OCO-2/3 and GOSAT achieve only sparse coverage in this challenging terrain. This dual limitation creates a fundamental challenge: how to quantify $CO_2$ emissions when available satellite data are either incomplete or spatially constrained. Here, the COCCON instrument offers a key advantage by simultaneously probing both columnar CO and

CO₂ concentrations, enabling a more direct linkage between CO and CO₂ emissions and helping to bridge the current
observational divide.
To address these limitations and quantify total emissions from the Xining area, we deployed an EM27/SUN FTIR
spectrometer in Ganhe Industrial Park (36.546°N, 101.518°E, 2603 m a.s.l.) during May-June 2024. The site was
strategically positioned in the southwest suburbs to capture integrated urban plumes downwind of the city center under
prevailing easterly winds. The early summer measurement period minimizes contributions from seasonal heating
emissions—particularly significant CO sources from combustion during colder months—thereby focusing on
emissions more characteristic of persistent urban sources. These ground-based measurements are complemented by
co-located TROPOMI and IASI satellite observations and CAMS model forecasts. Furthermore, we apply a simple
dispersion model to estimate CO and CO₂ emission rates from the urban area.

## 2. Methods and materials

### 2.1 Site description

A three-week field campaign was conducted in Ganhe Industrial Park (36.546°N, 101.518°E, 2603 m a.s.l.) from May
23 to June 14, 2024. The industrial park is located southwest of Xining, the capital of Qinghai Province, China (Figure
1) and was established in 2002 in the Huangzhong District. Xining, situated on the eastern edge of the Tibetan Plateau
and upstream of the Huangshui River, experiences a plateau mountain climate. In 2020, the city has a resident
population of approximately 2.47 million, with a density of 324 persons/km² (Xining Statistical Bureau, 2021), making
it the most densely populated area in Qinghai Province with nearly 80% of its population living in the urban center.

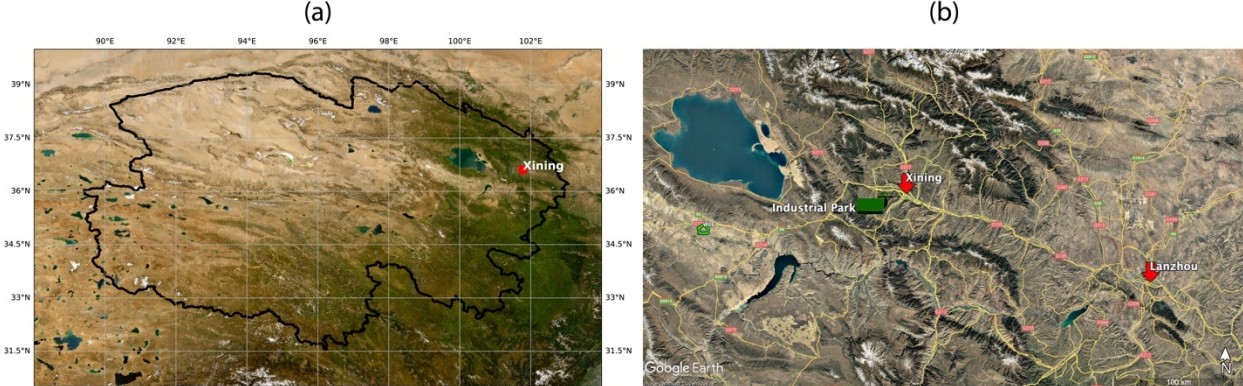


**Figure 1: A terrain map showing Qinghai Province and the location of its capital city of Xining (Terrain information originates from World Imagery). A map illustrating the locations of the EM27/SUN instrument within the Ganhe Industrial Park, Xining city (Qinghai province), Lanzhou city (Gansu province), and Waliguan station. The base map is sourced from © Google Earth, Image © 2024 Maxar Technologies; Image © 2024 CNES / Airbus.**

### 2.2 COCCON GHG products

In this study, a portable ground-based Fourier-transform infrared (FTIR) spectrometer, the EM27/SUN, was used.
Developed by the Karlsruhe Institute of Technology (KIT) in collaboration with Bruker Optics GmbH, Ettlingen,
Germany, the EM27/SUN is designed to measure solar absorption spectra in the near-infrared spectral range (covering
5500–11,000 cm$^{-1}$ and 4000–5500 cm$^{-1}$) with a spectral resolution of 0.5 cm$^{-1}$ (Gisi et al., 2012). The spectrometer
records double-sided, DC-coupled interferograms using two indium gallium arsenide (InGaAs) detectors at room
temperature (Hase et al., 2016). These interferograms are processed using a preprocessing tool and the PROFFAST
nonlinear least squares fitting algorithm, which was developed by KIT in the framework of the COCCON-
PROCEEDS project funded by the European Space Agency (ESA) (Alberti et al., 2022; Herkommer et al., 2024).
This approach enables the retrieval of atmospheric concentrations of trace gases such as $O_2$, $CO_2$, $CH_4$, CO and $H_2O$.
The gas column is then converted to column-averaged dry-air mole fractions of the gas (Xgas) by dividing the gas
column by the co-observed $O_2$ column, using the well-known mole fraction of $O_2$ in dry air.
With meanwhile about 200 EM27/SUN instruments deployed worldwide, these spectrometers are widely used for
monitoring GHG concentrations and air pollutant, for estimating emissions from various sources, and for validating
satellite measurements (Chen et al., 2020; Frey et al., 2015; Hase et al., 2015; Herkommer et al., 2024; Luther et al.,
2019; Tu et al., 2021, 2020). Due to its excellent level of robustness and reliability, the EM27/SUN instrument can be
applied in both field campaigns and long-term deployment at fixed sites. These observations complement TCCON at
various locations enhances global GHG monitoring efforts. In this context, the COCCON was established to further
advance and standardize these observational efforts.
The specific EM27/SUN instrument used in this campaign operates within the framework of COCCON. Prior to
deployment, the spectrometer was calibrated against the TCCON reference instrument at KIT to establish instrument-
specific calibration factors for each target gas, ensuring consistency with the TCCON data products (Frey et al., 2019).
To guarantee data quality and monitor instrumental stability throughout the campaign, a comprehensive quality control
(QC) and quality assurance (QA) protocol was implemented, following established COCCON procedures. This
protocol focuses on two key parameters: the Instrumental Line Shape (ILS) and the XAIR ratio. The ILS,
characterizing the spectrometer's spectral response function, was determined through laboratory measurements using
a nonlinear least-squares fitting algorithm (Frey et al., 2015; Hase et al., 1999). An ILS measurement performed in
December 2024 revealed a minor change (a ~1.5% decrease in modulation efficiency (ME)) compared to the pre-
campaign baseline at KIT, which is within the accepted uncertainty range and attributable to transportation. This small
change is attributed primarily to mechanical stresses incurred during the long-distance shipment from Germany to
China and subsequent domestic transports. A potential minor contribution from systematic errors associated with the
specific light source and lens used in the Chinese ILS setup cannot be entirely ruled out. The stability of the ILS was
confirmed by a subsequent measurement in September 2025 as no change was found in the ME value.
The XAIR ratio, derived from the measured vertical columns of $O_2$ and $H_2O$ alongside surface pressure, served as an
independent indicator of instrumental performance. The mean XAIR value during the campaign was $1.0012 \pm 0.0024$,
with nearly identical values recorded before (1.0005) and after (1.0005) the measurement period, demonstrating
excellent instrumental stability. Furthermore, to minimize uncertainties related to air mass dependence, a strict solar
zenith angle (SZA) filter of <60º was applied to the data. The combination of rigorous calibration with respect to
TCCON, continuous stability monitoring via ILS and XAIR, and careful data filtering ensures the high quality and
reliability of the ground-based data, providing a solid foundation for the subsequent satellite validation exercise.

## 2.3 TROPOMI CH$_4$ and CO products

The TROPOspheric Monitoring Instrument (TROPOMI), single payload of the Sentinel-5 Precursor (S5P) satellite, has been in orbit since October 2017. It operates in a low Earth polar orbit with a planned operational lifespan of 7 years. S5P is the first Copernicus mission and is designed to deliver daily global information on concentrations of traces gases (e.g., CH$_4$ and CO) and aerosols, aiming to monitor air quality, climate forcing and ozone abundances with high spatial and temporal resolution (Veefkind et al., 2012).

TROPOMI, currently the most advanced nadir-viewing and multispectral imaging spectrometer, was developed jointly by the European Space Agency (ESA) and the Netherlands Space Office. It measures across several spectral bands: ultraviolet (UV) and visible (VIS) (270–500 nm), near-infrared (NIR, 675–775 nm) and shortwave infrared (SWIR, 2305–2385 nm). With a wide swatch of 2600 km across track, TROPOMI provides operational level 2 (L2) CH$_4$ and CO products with a very high spatial resolution of approximately 5.5 km × 7 km since August 2019.

The retrieval of TROPOMI total column abundances of CH$_4$ is conducted using the RemoTec-S5P algorithm (Butz et al., 2009; Hasekamp and Butz, 2008). For CO, the total column retrieval is performed by a modified SWIR CO retrieval (SICOR) algorithm, which is based on the CO absorption band between 2305 nm and 2385 nm, with interfering trace gases and effective cloud parameters (Landgraf et al., 2016).

In this study, the TROPOMI L2 CH$_4$ and CO data with quality assurance values (qa_value) greater than 0.5, as recommend in the S5P product readme files, are utilized. Detailed documentation can be accessed for the CH$_4$ (https://sentinel.esa.int/documents/247904/3541451/Sentinel-5P-Methane-ProductReadme-File, last accessed: June 9, 2025) and CO data products (https://sentinel.esa.int/documents/247904/3541451/Sentinel-5P-CarbonMonoxide-Level-2-Product-Readme-File, last accessed: June 9, 2025).

## 2.4 IASI CO products

The Infrared Atmospheric Sounding Interferometer (IASI) is the primary payload carried on the EUMETSAT's MetOp series of polar-orbiting satellites and delivers meteorological parameters (e.g., water vapour and atmospheric temperature), and trace species with an unprecedented spatial and temporal coverage (Clerbaux et al., 2009). IASI sensors are nadir looking thermal infrared sensors and there are currently three instruments in operation, which were launched in 2006, 2012 and 2018.

The IASI CO dataset is processed using the Fast Optimal Retrievals on Layers for IASI (FORLI) software (George et al., 2009; Hurtmans et al., 2012). The retrievals are performed within the 2143-2181.25 cm$^{-1}$ spectral range based on the optimal estimation theory and tabulated absorption cross sections at various pressures and temperatures to enhance the efficiency of the radiative transfer calculation. The instrument provides global coverage with a 2200 km swath and a 12 km resolution at nadir (George et al., 2009). In this study, total column abundances of CO from IASI onboard Metop-C Level 2 (version 6.7) are used.

**2.5 CAMS CAMS high-resolution GHG forecasts and CAMS-GLOB-ANT inventory**

The Copernicus Atmosphere Monitoring Service (CAMS), which is implemented by the European Centre for Medium-Range Weather Forecasts (ECMWF), produces daily global forecasts for the two main long-lived GHGs. As part of the CAMS GHG services, the ECMWF Integrated Forecasting System (IFS) delivers 5-day high-resolution forecasts of $CO_2$, $CH_4$ (Agustí-Panareda et al., 2014, 2017, 2019), as well as CO and meteorological parameters essential for GHG forecasting (Flemming et al., 2015). The forecast is generated a few hours behind real time, with initial conditions derived from a 4-day forecast of the analysis experiment. It is run at a horizontal resolution TCo1279, corresponding to a cubic octahedral reduced Gaussian grid with an approximately spatial resolution of 9 km (Holm et al., 2016). The model includes 137 vertical levels extending from the surface to 0.01 hPa (Agustí-Panareda et al., 2019).

In this study, the forecasting suite is based on the IFS model cycle CY48R1, which was upgraded in June 2023. This update introduced several system enhancements, particularly in composition modelling, emissions and assimilation (Eskes et al., 2024). Notably, CY48R1 includes the assimilation of TROPOMI CO and the performance has generally improved compared to all observations, such as surface observations, vertical profiles from IAGOS aircraft and NDACC FTIR measurements, and satellite total column retrievals (Eskes et al., 2024).

CAMS also delivers global anthropogenic emissions, referred to as CAMS-GLOB-ANT. This includes emissions of both air pollutants (e.g., CO) and greenhouse gases (e.g., $CO_2$ and $CH_4$) for real-time forecasts (Soulie et al., 2024). Emissions are provided for 17 sectors and 35 species as monthly averages, with a spatial resolution of $0.1° \times 0.1°$, covering the period from 2000 to 2025. This study uses the latest version v6.2, which is based on the Emissions Database for Global Atmospheric Research (EDGARv6) and employs the same methodology as v5.3.

**2.6 Dispersion model and wind-assigned anomaly method**

For a single point source, the total emission is calculated by multiplying the measured total column enhancement (ΔCO) by the area of the affected plume (Babenhauserheide et al., 2020). This plume area is modeled as an evenly distributed cone, representing the long-term averaged dispersion (Tu et al., 2022a). The relationship is given by the following equation:

$$\varepsilon = \Delta CO \times d \times v \times \partial \qquad \textbf{Eq. 1}$$

where ΔCO represents the enhanced CO column observed at the downwind site, $d$ is the distance from the source to the measuring site and $v$ is the wind speed.

To estimated averaged emissions from satellite observations over a region, the wind-assigned method was applied (Tu et al., 2022a, 2022b, 2023, 2024b). This technique fits the anomalies between the satellite observations and the dispersion model by analyzing enhancements under opposing wind sectors. Specifically, the wind-assigned anomaly is defined as the difference in observed enhancements between two opposite wind fields (e.g., E: 0°–180° and W: 180°–360°). A key advantage of this approach is that it inherently eliminates the uncertainty associated with

background concentration calculations for long-lived gases like CO, thereby significantly improving the reliability of
the resulting emission estimates.

## 3. Results and discussion

### 3.1 Ground-based observations

Figure 2 illustrates the time-series of $XCO_2$, $XCH_4$ and XCO observed by the EM27/SUN over 8 intermittent days
from May 25 to June 14. $XCO_2$ show a mean value of $426.50 \pm 1.79$ ppm with relatively less intraday variability
compared to CO. A generally lower concentration was observed on June 4 ($424.64 \pm 0.25$ ppm), except for a slight
enhancement (~1.2 ppm) in the morning, which was also observed in $XCH_4$ (~6 ppb) and in XCO (~15 ppb).
The average observed $XCH_4$ concentration during the measurement period was $1898.45 \pm 6.66$ ppb. Notably, 300 km
to the northwest of the study site lies the largest Muli coalfield in Qinghai Province, which has an estimated coal
reserve of around 4 billion tons (Xiao et al., 2023). However, the maximum $XCH_4$ enhancement observed was
approximately 10 ppb on May 25, when the wind was from northwest before noon. This enhancement is relatively
lower than those reported in other coal fields, such as in Changzhi, Shanxi Province (Tu et al., 2024a), suggesting that
the ground-based observations may not have fully captured the methane emissions. The modest enhancement is likely
due to the relatively greater distance from the source, combined with potentially low coal mining activity during the
observing period, as coal production is often highly episodic.
XCO exhibited a mean value of 153.75 ppb and a standard deviation of 52.09 ppb. These values are significantly
higher than those in the Arctic and Antarctica (Pollard et al., 2022; Sha et al., 2024), as well as in urban areas such as
Seoul (Park et al., 2024), Xianghe near Beijing (Che et al., 2022a; Yang et al., 2020) and Hefei (Shan et al., 2022).
Among the investigated gases, XCO exhibits the most significant intraday variability, particularly on May 25, 27 and
June 14. During these days, prevailing easterly winds transported airmasses from Xining City and the nearby highway
(Xiong et al., 2022), leading to the elevated XCO levels. In contrast, on June 4, when airmasses primarily originated
from the west and northwest—regions with less urbanized areas—XCO levels were notably stable and low, averaging
$93.93 \pm 4.23$ ppb (see back trajectories on Figure A 2). Additionally, higher wind speed on June 4 facilitated the
dispersion of CO, contributing to the lower CO level. The enhancement of XCO and $XCO_2$ ratio ($\Delta XCO:\Delta XCO_2$, see
section 3.5) exhibited slopes of 14.43 ppb/ppm before noon and 4.76 ppb/ppm in the late afternoon. Both values were
significantly lower than those observed under easterly wind conditions. This suggests that the CO and $CO_2$ emissions
in the western regions originate from different combustion processes or source types compared to those in the east.

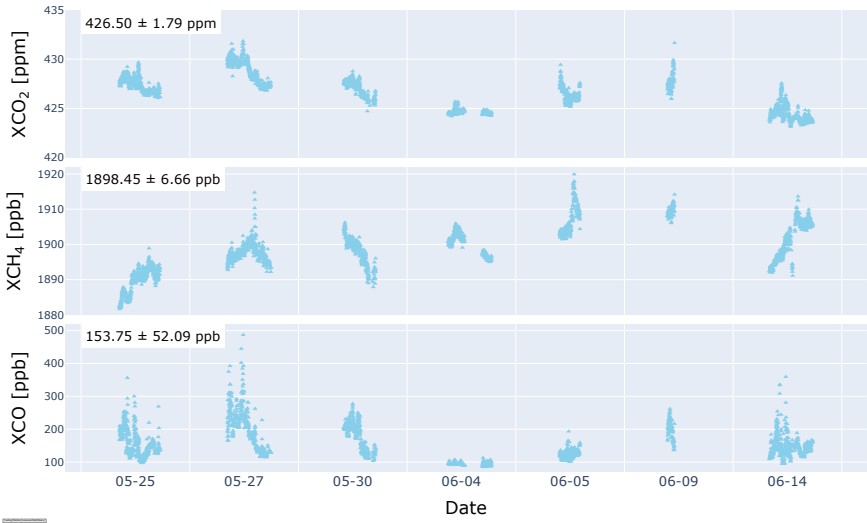

**Figure 2: time-series of EM27/SUN observations from May 25 to June 14 2024.**

**3.2 Comparison of COCCON data with TROPOMI and IASI products**

Figure 3a-b illustrates the correlations between TROPOMI and COCCON measurements on different days. The spatial collocation criterion requires the TROPOMI observations to fall within a radius of 200 km for $XCH_4$ and 100 km for $XCO$. The temporal collocation criterion is set to ±2h for COCCON measurements.

The results indicate that TROPOMI tends to slightly overestimate $XCH_4$ by an average bias of 8.74± 12.19 ppb, while it significantly underestimates of $XCO$, showing a bias of 49.49 ± 25.76 ppb. The correlation between TROPOMI and COCCON $XCH_4$ measurements improves as the spatial distance between their respective locations decreases. The largest observed difference in $XCH_4$, approximately -20.66 ppb, occurs on June 14, when the minimal distance between TROPOMI and COCCON locations is around 150 km (Figure A 3a).

A similar trend is observed for $XCO$, where larger biases are generally associated with greater distances. However, on May 27, the largest bias (69.23 ppb) occurs despite a relatively small minimum distance of 13 km. This significant discrepancy is likely influenced by the spatial distribution of TROPOMI observations relative to the EM27/SUN location. Notably, on May 27, TROPOMI $XCO$ levels exhibit a negative correlation with distance for observations within approximately 100 km (Figure A 3b). The closest observation to the EM7/SUN location records a TROPOMI $XCO$ value of 157.20 ppb, resulting in a significantly reduced bias of 16.72 ppb.

CO emissions from vehicle exhaust, a major contributor to air pollution, is closely related to fossil fuel combustion (Naus et al., 2018). Gao et al. (2025) reported that CO emissions increased significantly with altitude, observing nearly twice the emission levels in Xining (2320 m) compared to those at an altitude of 20 m. This trend can be attributed to the decline in atmospheric pressure and air density at higher elevations (Fattah et al., 2019). Under such conditions, engines draw in less air per cycle, which alters the air-fuel ratio and leads to suboptimal combustion. As altitude rises, the excess air ratio decreases because more diesel fuel is injected into the cylinders, but less air is captured per cycle. Therefore, the combustion between fresh air and diesel fuel becomes incomplete, resulting in substantial CO emissions.

The correlation between IASI and COCCON is relatively higher ($R^2$ = 0.7211) compared to the correlation between
TROPOMI and COCCON XCO. Similar to TROPOMI, IASI shows the highest bias on May 27 with a value of
2.43E22 molec./m$^2$. Enhanced CO levels are observed within 20 km radius, with the peak CO concentration reaching
2.66E22 molec./m$^2$ at 15.4 km (Figure A 3c). As a result, the bias decreases by nearly half to 1.32E22 molec./m$^2$.

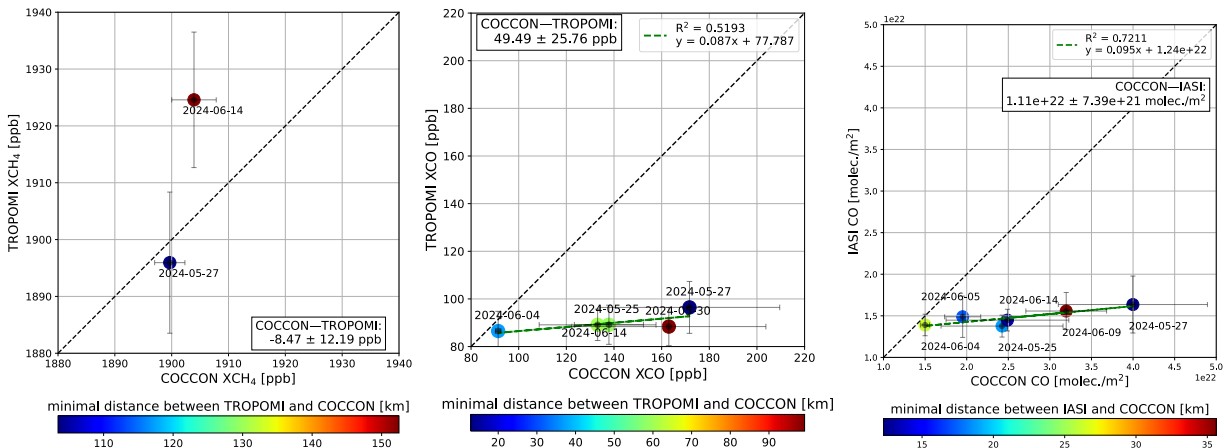


**Figure 3: Correlation plot between TROPOMI and COCCON for XCH$_4$ (a) and XCO (b) and between IASI and COCCON**
**for CO column (c). The colour bar represents the minimum distance between the TROPOMI and COCCON locations.**

## 3.3 Comparison of COCCON data with CAMS products

The CAMS GHG forecast provides a high spatial resolution of approximately 9 km. For comparison with the
EM27/SUN site, the following coincidence criteria were used: CAMS data nearest to the EM27/SUN location and
EM27/SUN observations within ± 2h around noon.
In daily comparison, the CAMS forecast XCH$_4$ demonstrates a strong correlation ($R^2$ = 0.7826) with COCCON
observations and an average bias of 4.21 ± 4.40 ppb. In contrast, CAMS generally underestimates both XCO$_2$ and the
CO column, exhibiting lower correlations. The mean biases are 1.78 ± 1.47 ppm for XCO$_2$ and 9.97E21 ± 6.31E21
molec./m$^2$ for CO. The largest biases in XCO$_2$ and CO are recorded on May 27. This discrepancy is also observed in
TROPOMI and IASI CO observations. When comprising the absolute column amounts, CAMS shows an
approximately 0.4% underestimation for XCO$_2$ and 0.2% for XCH$_4$ relative to the COCCON data. However, the
underestimate of CO is more pronounced, with a bias of 35%.
CAMS XCH$_4$ compares much better with COCCON observations than XCO$_2$ and XCO, as the higher value of $R^2$ and
the better agreement of the slope of the regression line indicates. We conclude that the variability of XCO and XCO$_2$
dominating the variability as detected by the ground-based observation is generated on a smaller spatial scale which
is not properly resolved in the simulation. In contrast, the variability of the CH$_4$ model field seems to be dominated
by extended sources distributed in a wider area, which therefore can be properly depicted by the model.
In terms of broader implications, the relatively strong performance of CAMS in simulating CH$_4$, especially when
compared to CO and CO$_2$, indicates that CAMS may be more reliable for CH$_4$ studies in such regions. This finding
could potentially support the case for establishing long-term observation sites in this area to help with satellite
validation and improve model accuracy, particularly for species like CO, which appear to require more refined
simulations.

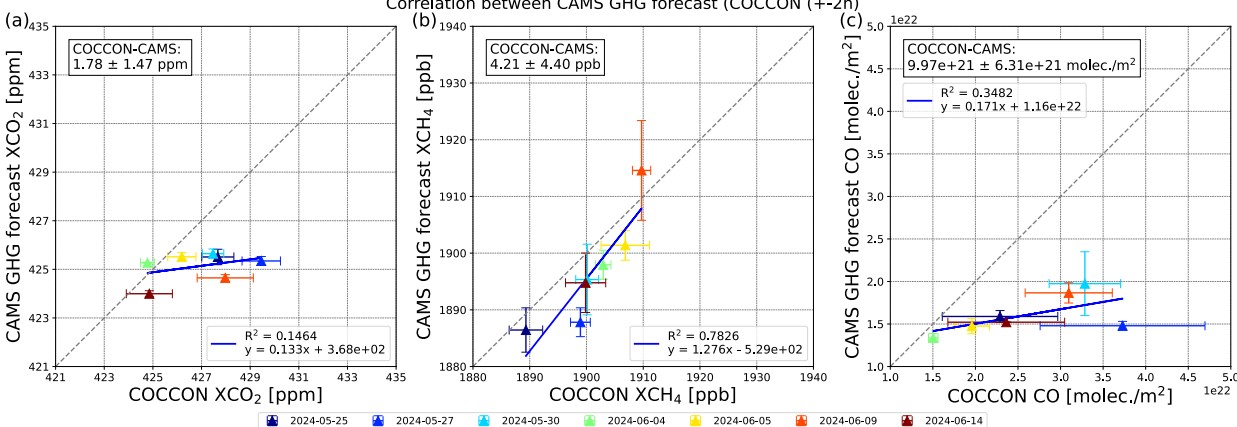


**Figure 4: Correlation between CAMS forecast and COCCON for XCO₂ (a), XCH₄ (b) and CO column (c).**

**3.3 CO emission estimates based on ground-based observations**

Satellite observations, such as TROPOMI and IASI, and CAMS forecasts, consistently underestimate CO levels
during the entire field campaign. Thus, ground-based measurements are essential for accurately estimating CO
emissions. Wind directions predominantly originated from the east (Figure A 1) on five days, indicating that emissions
from the Xining city were being transported to the downwind site where the EM27/SUN located.
Figure 5 highlights the correlation between ΔXCO multiplied by wind speed as a function of wind direction. ΔXCO
represents the emitted signals by subtracting backgrounds from the observations. The daily background is defined as
the lowest 10th percentile of measurements. Wind directions within the range of 80° – 120°, associated with relatively
higher multiplication between ΔXCO and wind speed, were selected to represent the predominant winds transporting
emissions from Xining. In this range, the average wind speed is approximately 2.3 m/s, with a standard deviation of
0.7 m/s. To account for variability, an uncertainty of ±20º is applied. The wind spreading angle $\partial$ is defined as:

$$\partial = 120° − 80° = 40°(\pm20°) = 0.7\ (\pm0.35)\ rad$$

Based on this wind spreading angle, the CO emission is calculated using the dispersion model in Equation 1. The
elevated values between 250° – 300° are mainly attributable to higher wind speeds. This sector is not the focus of the
present study, as the ground-based EM27/SUN instrument primarily observes air masses from Xining transported by
easterly winds.

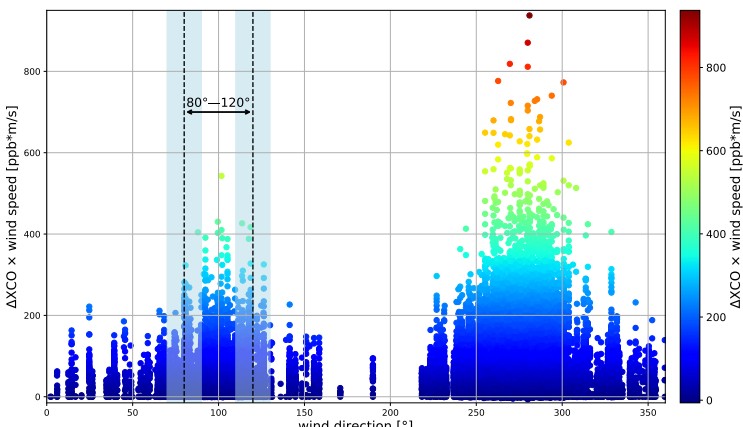


**Figure 5: TROPOMI ΔXCO multiplied by wind speed against the wind direction for wind speed greater than 1.5 m/s, covering the period from May 2018 to June 2024. Wind data are derived from ERA5 at a heigh of 100m above ground level at the EM27/SUN station at 14:00 local time. The ΔXCO dataset represents XCO residuals after background subtraction. The wind direction range of 80°–120° (delineated by dashed lines) captures the predominant wind influence from Xining city. The shaded areas represent the uncertainty of ±10°.**

To minimize accidental bias, a 10-minuten averaged EM27/SUN CO dataset was used. Significant enhancements are observed in ΔXCO and $ΔXCO_2$ in the 80°-120° wind direction range (Figure A 4). Using Equation (2), the estimated CO emissions for five days are presented in Figure 7. Notably, relatively high emissions were observed in the 80°-120° wind direction range, corresponding to areas of higher wind speed. A similar enhancement of CO associated with easterly winds was reported by Xiong et al. (2022), where in situ observations at the Waliguan station (36.28°N, 100.09°E, 3816 m a.s.l.), located approximately 155 km southwest of Xining city. This highlights the influence of regional air transport on the observed CO concentrations.

The CO emissions in the wind direction interval of 80°-120° are likely attributed to Xining city, with a mean estimated emission rate of 2.7E26 ± 2.1E26 molec./s (i.e., 12.3 ± 9.6 kg/s, see Figure 6). The maximum emission rate reached 1.2E27 molec./s (i.e., 55.6 kg/s) on May 27, a day when both satellite observations and forecasts underestimated the CO levels. The peak emission on May 27 occurred around noon, coinciding with a significant enhancement in XCO, which reached an approximate 10-minute averaged signal of 300 ppb.

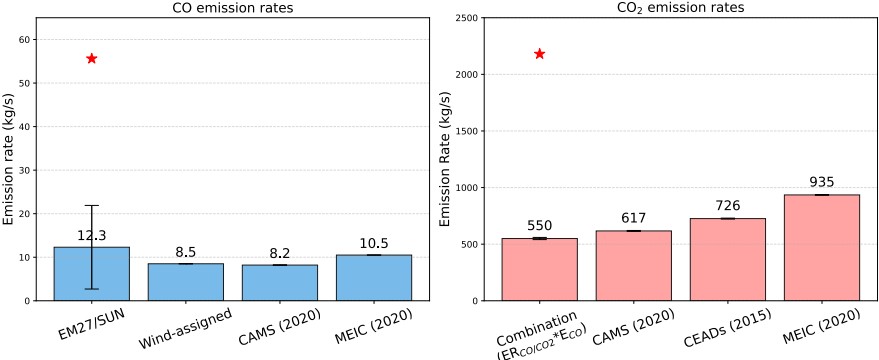


**Figure 6: CO and CO₂ emissions from this study and different inventories over Xining city. The red start symbols represent the highest value derived from EM27/SUN observations.**

Another notable high emission event was observed on May 25 and 30, under wind directions around 60°. These
elevated emissions are likely to attributed to emission from the north-northeast region of the EM27/SUN location. On
May 25, an enhanced XCO signal (~80 ppb) was observed in the late afternoon, coinciding with a relatively higher
wind speed of ~2.8 m/s, which contributed to the increased emissions. On March 30, the XCO signal peaked at ~120
ppb around noon, leading to even higher emission rates.

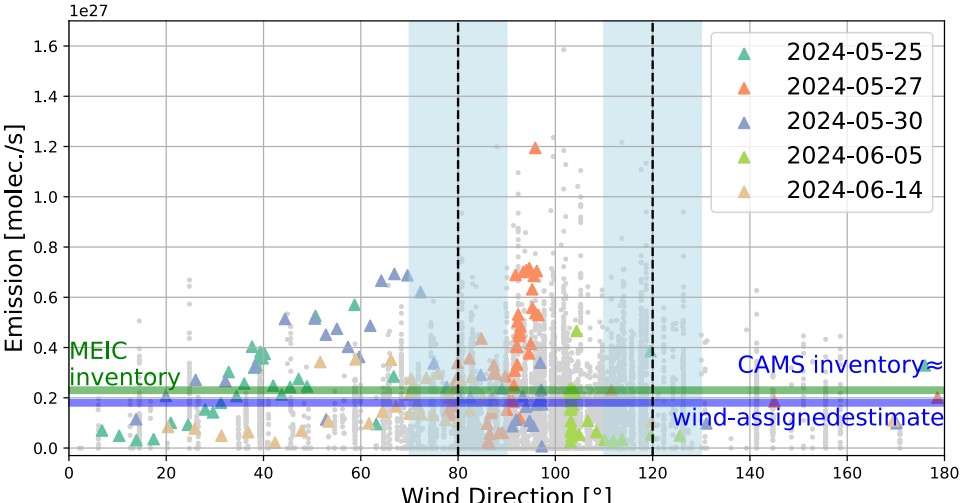


**Figure 7: Estimated emission relative to wind direction on six days of ground-based measurements (triangle symbols) when winds were predominantly easterly. Grey dots represent emissions derived from TROPOMI XCO. The green horizontal line indicates CO emission rates derived from the MEIC inventory, while the blue line represents both CAMS-GLOB-ANT inventory and wind-assigned anomaly method estimates, which yield nearly identical values.**

**3.4 CO emission estimates based on satellite-based observations**

TROPOMI observes relative high CO levels near the central region of Xining. A distinct streak of elevated CO
concentrations extends eastward from Xining toward the Qinghai province border and continues toward Lanzhou in
Gansu province (not shown in the figure). This pattern aligns with the location of densely populated residential regions
and coincides with a gradual decrease in altitude from west to east (Figure A 5).

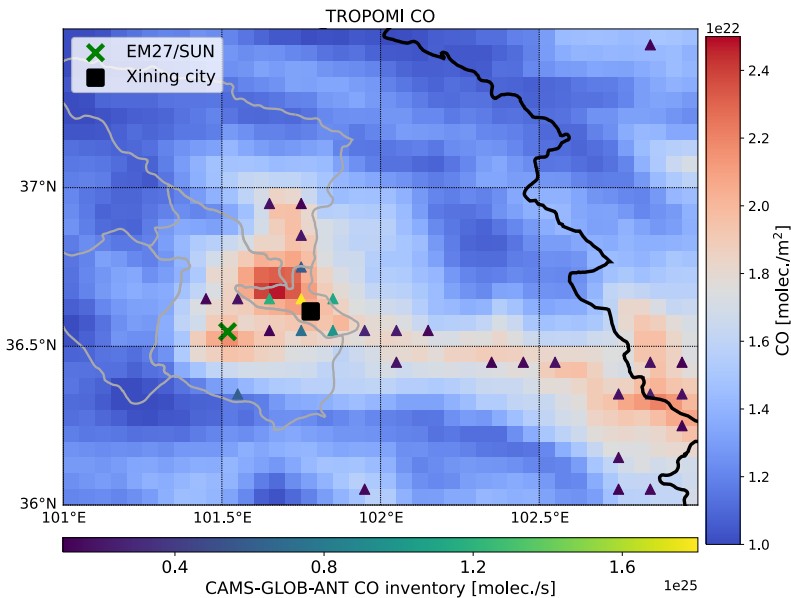


**Figure 8: Spatial distribution of the average TROPOMI CO column at a latitude-longitude resolution of 0.05º, covering the period from May 2018 to June 2024. The triangle symbols indicate locations where CO emissions exceed 1E24 molec./s based on the CAMS-GLOB-ANT inventory. The green cross marks the location of the EM27/SUN and the black square represents the center of Xining city. The thick black line outlines the Qinghai province boundary and the thin grey lines delineate subregions within Xining.**

Emissions derived from TROPOMI CO products, based on the Eq1. are presented in Figure 7 (grey dots), showing relatively higher values for wind directions between 80º and 120º. The average emission rate is approximately 8.9 ± 7.5 kg/s, which closely matches the emissions derived from the COCCON measurements.

To further estimate regional CO emissions, a wind-assigned method in combination with the dispersion model described in Eq.1 is applied to the TROPOMI observations. The estimated CO emission based on the wind-assigned anomaly method is approximately 1.8E26 molec./s (8.5 kg/s). This value is comparable to the CAMS-GLOB-ANT inventory (1.8E26 molec./s, i.e., 8.2 kg/s), the Multi-resolution Emission Inventory model for Climate and air pollution research (MEIC, http://meicmodel.org.cn, last access: 1 September 2025), (Geng et al., 2024; Li et al., 2017), which reports 10.5 kg/s, and the average emission derived from the six-day EM27/SUN measurements (2.7E26 molec./s, i.e., 12.3 kg/s). However, scatters are observed in the correlation between the TROPOMI and modeled wind-assigned anomalies, leading to a lower $R^2$ value (Figure 9). This discrepancy could originate from uncertainties in the CAMS inventory, which may not account for some some-high emission sources. Additionally, the simple dispersion model introduces its own uncertainties, which limits the ability to accurately model CO enhancements.

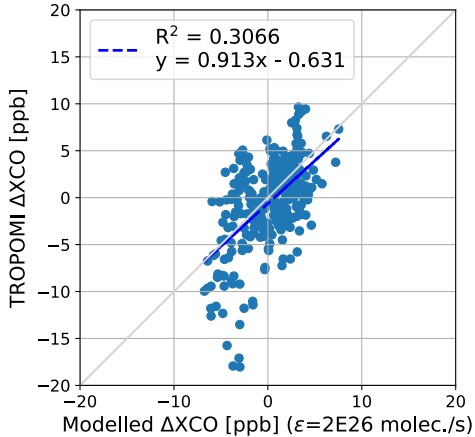

352

**Figure 9: Correlation in the wind-assigned anomalies between the TROPOMI observations and modeled anomalies.**

**3.5 CO₂ estimates based on combustion efficiency**

In urban environments, CO is primarily produced through the incomplete combustion of fossil fuels, such as those from traffic emissions and industrial activities, often occurring alongside the formation of $CO_2$. The CO to $CO_2$ ratio serves as a valuable indicator for atmospheric fossil-fuel $CO_2$ emissions and combustion efficiency, providing insights into the sources and effectiveness of fuel use (Che et al., 2022b; Lee et al., 2024; Sim et al., 2020).

Figure 10 presents the relationship between ΔXCO and ΔXCO₂ under predominant wind of wind direction of 80° – 120° and wind speed larger than 1.5 m/s for different days. The daily background concentrations of $XCO_2$ and XCO were determined as the lowest 10th percentile of the respective observations. ΔXCO and ΔXCO₂ was calculated by subtracting the daily background concentrations from the daily observed values. Under easterly wind conditions, ΔXCO exhibits a stronger correlation with ΔXCO₂, with a slope of 35.14 (ppb/ppm) and an $R^2$ value of 0.7552, indicating distinct source contributions and atmospheric transport processes from eastern regions, such as Xining city. The observed ΔXCO/ΔXCO₂ ratio is higher than those reported in other studies (Che et al., 2022b; Lee et al., 2024). This discrepancy is attributed to significantly elevated CO levels under easterly wind conditions, suggesting lower combustion efficiency in this region.

Using CO as a proxy, the fossil fuel-derived $CO_2$ emissions can be computed as follows:

$$E_{CO_2} = (\frac{1}{ER_{CO/CO_2}} \times \frac{M_{CO_2}}{M_{CO}})E_{CO}$$

**Eq. 2**

Where $E_{CO_2}$ is the emissions of $CO_2$ in kg/s, $ER_{CO/CO_2}$ is the ground-based observed ratio of ΔXCO:ΔXCO₂ in ppb/ppm, $E_{CO}$ is the CO emission rate derived from TROPOMI observations (see Sec. 3.4), and $M_{CO_2}$ and $M_{CO}$ are the molar mass of $CO_2$ and CO, respectively.

We estimate an average $CO_2$ emission rate of approximately 550 kg/s, which aligns well with the CAMS-GLOB-ANT (617 kg/s for 2020) though lower than the Carbon Emission Accounts and Datasets (CEADs) (726 kg/s for 2015) ("Methodology and applications of city level CO2 emission accounts in China," 2017; Shan et al., 2018) and MEIC

(935 kg/s for 2020) estimates. The data also reveal strong daily fluctuations in emissions. The peak event was observed
on May 27, which exhibited a maximum $\Delta XCO:\Delta XCO_2$ ratio of 40.08 ($R^2 = 0.8544$). This ratio translates to a
maximum CO emission rate of 55.6 kg/s and a concurrent maximum $CO_2$ emission rate of 2180 kg/s.
Additionally, the CAMS and MEIC inventories show similar $CO/CO_2$ emission ratios of 0.021 and 0.018, respectively.
As detailed in Section 3.2 and 3.3, both TROPOMI and CAMS model underestimate the atmospheric CO column by
a factor of approximately 1.6. When we correct for this bias by scaling the TROPOMI-derived emission and CAMS
inventory, the resulting emission ratio increases to 0.034. This corrected value aligns closely with our observed
$\Delta XCO/\Delta XCO_2$ enhancement ratio of 0.035 ppb/ppb.
The observed discrepancies compared with inventories may be attributed to differences in temporal coverage,
methodological approaches, and potential changes in emission patterns over time. Additionally, it should be noted that
the field campaign spanned only three weeks from May to June, which mainly represents early summer. During other
seasons, such as summer or winter, when photosynthesis activities or coal burning for heating is more prevalent, the
$\Delta XCO:\Delta XCO_2$ ratios and associated $CO_2$ emissions may differ. A longer period of ground-based observations and
running several spectrometers upwind and downwind may improve our results. Our findings so far demonstrate the
potential of the EM27/SUN spectrometer as a promising tool for comprehensively evaluating greenhouse gas (GHG)
and air pollutant emissions in urban areas (Che et al., 2022b; Lee et al., 2024).

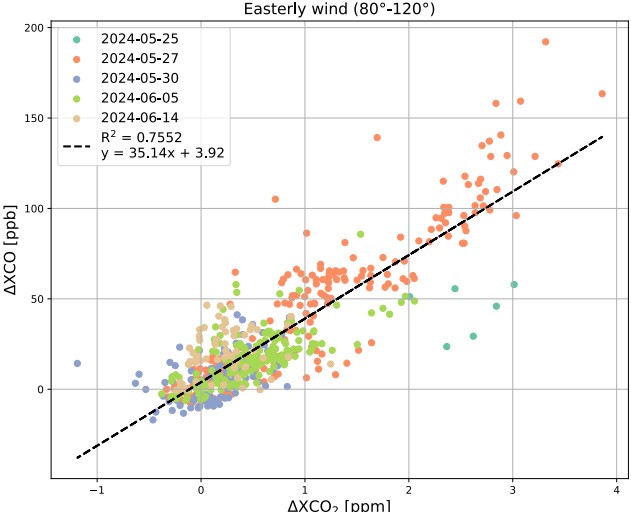


**Figure 10: Ground-based observed $\Delta XCO/\Delta XCO_2$ ratios under easterly wind conditions covering direction between 80°**
**and 120°. Different colors indicate observations from individual days.**
**Conclusion**
A three-week field campaign using a portal FTIR spectrometer (EM27/SUN) was conducted at Ganhe Industrial Park,
located in the southwestern suburbs of Xining city, from May 23 to June 14, 2024. The mean and standard deviation
values for $XCO_2$, $XCH_4$ and XCO were 426.50±1.79 ppm, 1898.45±6.66 ppb and 153.75±52.09 ppb, respectively.
Among these gases, XCO exhibited significant intraday variability, particularly on days dominated by easterly winds,
which transported airmass from Xining city.

Ground-based observations were compared with co-located datasets from TROPOMI, IASI and CAMS. Results indicate that TROPOMI slightly overestimates $XCH_4$, with an average bias of $8.47 \pm 12.19$ ppb, but significantly underestimates XCO, showing a bias of $49.49 \pm 25.76$ ppb. IASI generally underestimates the CO column relative to COCCON observations, with an average bias of $1.12E22 \pm 8.09E21$ molec./m$^2$. CAMS forecasts also underestimated $XCO_2$ ($1.78 \pm 1.47$ ppm), $XCH_4$ ($4.21 \pm 4.40$ ppb) and CO column ($9.97E21 \pm 6.31E21$ molec./m$^2$). A high correlation ($R^2 = 0.7930$) was found for $XCH_4$ between CAMS and EM27/SUN datasets. The largest discrepancies in CO products occurred on May 27, when significant enhancements were observed at the co-located time by EM27/SUN. These discrepancies are partly attributed to satellite observations, as their concentrations decreased with distance from the EM27/SUN location.

Wind directions predominantly originated from the east on five measurement days, indicating that emissions from the Xining city were being transported to the downwind site where the EM27/SUN spectrometer was located. A simple dispersion model, incorporating wind information and enhanced CO column data from EM27/SUN instrument, was applied to estimate CO emissions. The CO emissions from Xining city are estimated at $12.3 \pm 9.6$ kg/s with a peak emission rate of 55.6 kg/s on May 27, when both satellite observations and forecasts underestimated CO levels. When this model was applied to TROPOMI CO data, the resulting average emission rate was $8.9 \pm 7.5$ kg/s. A wind-assigned anomaly method was also applied to the TROPOMI CO dataset, resulting in an estimate emission rate of 8.5 kg/s. Both EM27/SUN-based and TROPOMI-based estimates are comparable to the CAMS inventory value of 8.2 kg/s in 2020.

CO can serve as a proxy for fossil fuel-derived $CO_2$ emissions. Ground-based obversions of $\Delta XCO$ and $\Delta XCO_2$ exhibit a stronger correlation under wind direction of $80°-120°$, with a slope of 35.14 ppb/ppm and an $R^2$ value of 0.7552, compared to conditions under westerly winds. Using these correlations, we estimated the $CO_2$ emission rate to have an average value of 550 kg/s with a maximum value of 2180 kg/s. These estimates align reasonably with the CAMS-GLOB-ANT (617 kg/s for 2020) and the CEASs inventory, which reports a $CO_2$ emission rate of 726kg/s for Xining in 2015. Observed discrepancies may be attributed to differences in temporal coverage, methodological approaches, and potential changes in emission patterns over time. Note that long-term ground-based measurements of trace gases may improve the precision of estimated emissions.

Our findings highlight the potential of the EM27/SUN spectrometer as a valuable tool for detecting local emission and supporting satellite validation, particularly in high-altitude regions such as Tibetan Plateau. This study establishes a novel approach for estimating $CO_2$ emissions by synergistically combining ground-based and space-based measurements. While TROPOMI provides extensive spatial coverage for CO, the lack of direct $CO_2$ measurements from this platform—coupled with the sparse coverage of dedicated $CO_2$ satellites (OCO-2/3, GOSAT) in this region—highlights a critical observational gap. The simultaneous measurement of columnar CO and $CO_2$ by EM27/SUN provides a key advantage, enabling direct correlation of their emissions and offering a cost-effective approach for comprehensive greenhouse gas and air pollutant assessment.

**Data availability.**

The TROPOMI CH₄ and CO dataset is a Copernicus operational product and is available at https://doi.org/10.5270/S5P-3lcdqiv (Copernicus Sentinel-5P, 2021). The access and use of any Copernicus Sentinel data available through the Copernicus Open Access Hub are governed by the legal notice on the use of Copernicus Sentinel Data and Service Information, which is given at https://sentinels.copernicus.eu/documents/247904/690755/Sentinel_Data_Legal_Notice (last access: January 2025). IASI CO dataset is available at https://iasi.aeris-data.fr/co/ (last access: January 2025). CAMS forecasts are available at 10.24381/93910310 (last access: January 2025) and CAMS-GLOB-ANT inventory data are available at https://doi.org/10.24380/eets-qd81 (last access: January 2025). CEADS inventory is available at: https://www.ceads.net/data/city?#1280. last access: January 2024.

**Author contributions.**

QT and FH conceived the research question. QT wrote the manuscript and conducted the data analysis with input from FH. XL and ZY carried out the measurements, while YJ and JF was responsible for the data processing. All authors contributed to the interpretation of the results and the improvement of the paper.

**Declaration of Competing Interest.**

The authors declare that they have no known competing financial interests or personal relationships that could have appeared to influence the work reported in this paper.

**Acknowledgements.**

We would like to thank Copernicus User Support Team at ECMWF for providing the CAMS model data. We thank the TROPOMI team for making CH₄ and CO data and IASI team for CO data publicly available. We are grateful to the handling editor and two anonymous reviewers for their constructive comments that significantly improved this work.

**Financial support.**

This research was funded by the National Natural Science Foundation of China (grant no. 42305138) and the National Key Research and Development Program (2022YFE0209500).

**Appendix**

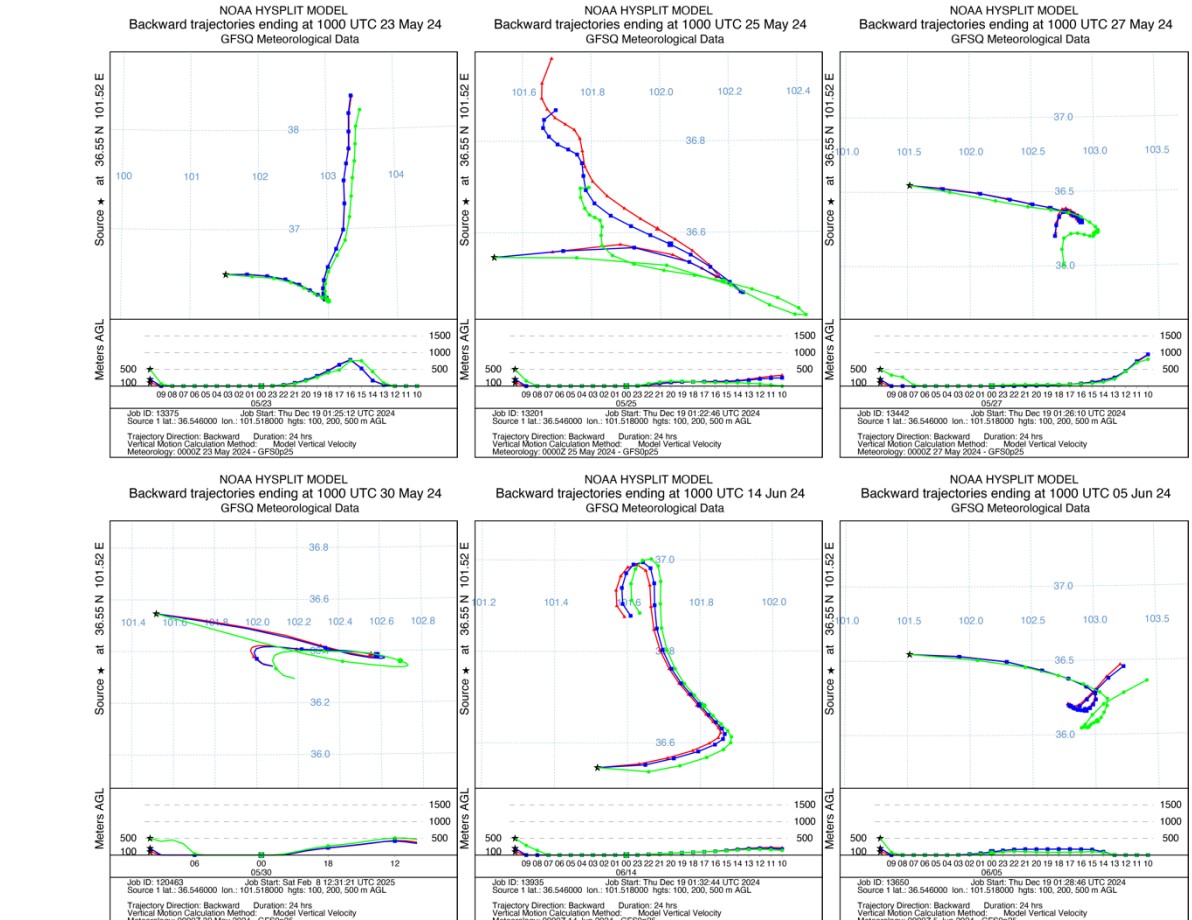

**Figure A 1 The back trajectories originating from eastern areas and arriving at EM27/SUN station at 10 UTC (18 Local**
**time) on different days. The trajectories, calculated using the HYSPLIT model (Hybrid Single-Particle Lagrangian**
**Integrated Trajectory), and shown for height of 100, 200 and 500 m above ground level.**

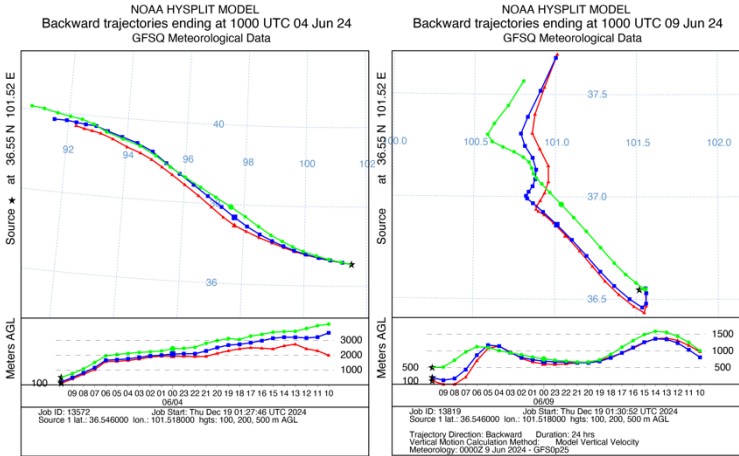

**Figure A 2 Similar to Figure A 1, but for trajectories generally from western areas on June 4 and 9, 2024.**

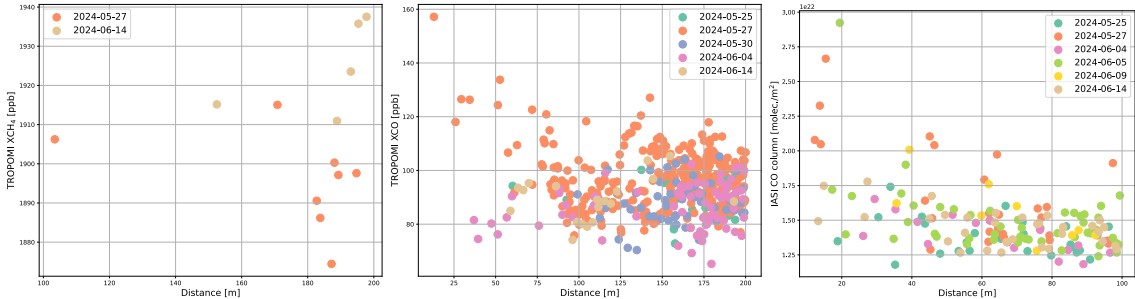


**Figure A 3: Correlation between TROPOMI XCH₄ (a) and XCO (b) with the distance from each TROPOMI observation to the EM27/SUN location. Panel (c) shows the correlation between IASI CO and the distance from each IASI observation to the EM27/SUN location.**

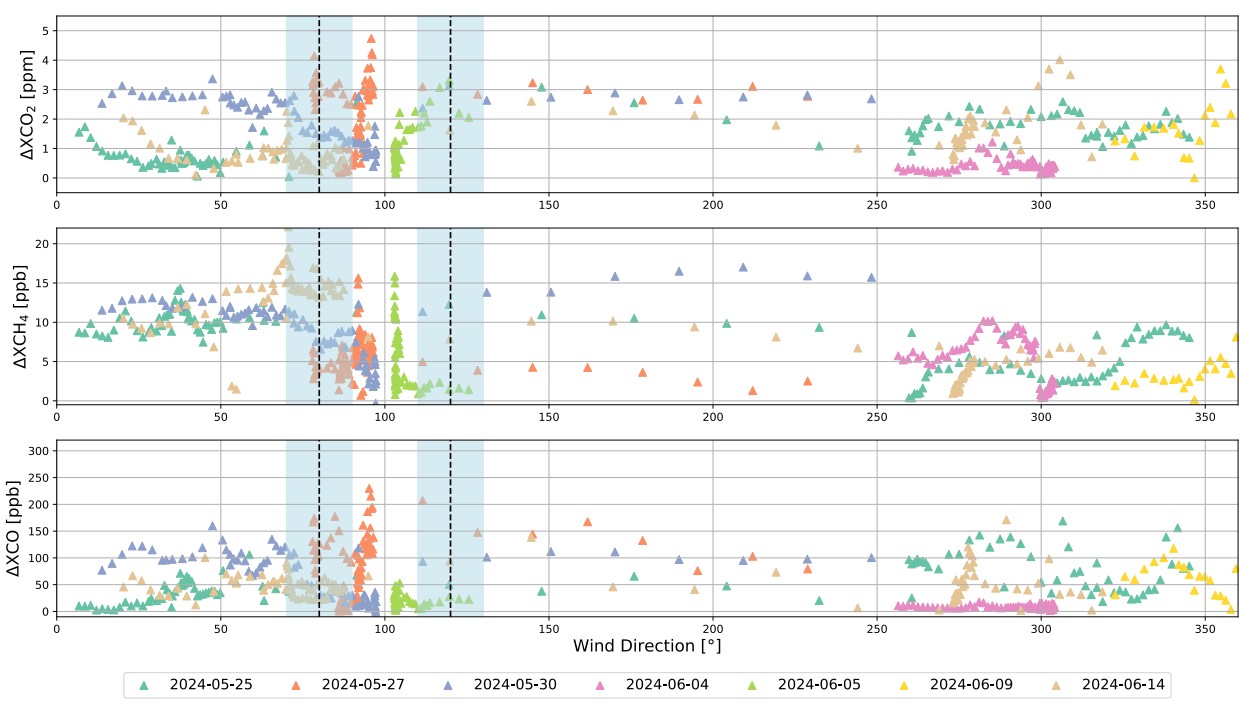


**Figure A 4: correlations between $\Delta XCO_2$, $\Delta XCH_4$ and $\Delta XCO$ with wind direction. The wind direction range of 80°-120° is delineated by dashed lines.**

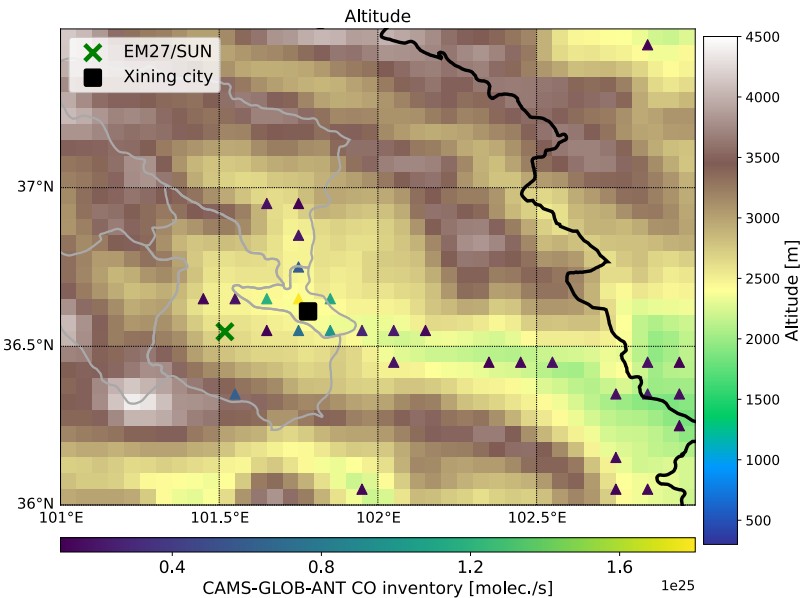


**Figure A 5: similar to Figure 8 but for terrain distribution.**

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
