# Peer review of "in a Qinghai-Tibetan Plateau city using a portable Fourier"

_EGUsphere, 2025_

## Author Comment (AC1)

**Response to Referee #1**

We would like to thank reviewer #1 for taking the time to review this manuscript and for providing valuable, constructive feedback and corresponding suggestions that helped us to further improve the manuscript.

In this author's comment, all the points raised by the reviewer are copied here one by one and shown in black color, along with the corresponding reply from the authors in blue.

This manuscript presents a short-term measurement campaign in Xining, a city located on the eastern edge of the Qinghai-Tibetan Plateau (QTP), using a portable FTIR instrument (EM27/SUN) to retrieve column-averaged concentrations of CO2, CH4, and CO. The study touches on several topics, including satellite validation for CH4 and CO, CAMS product evaluation, combustion efficiency derived from the CO:CO2 ratio, and CO2 emissions estimation. However, the manuscript lacks a cohesive narrative and frequently shifts between topics without adequately developing or concluding each one. As a result, it reads more like a collection of loosely connected sub-studies rather than a focused, hypothesis-driven investigation.

We are grateful to the referee for this insightful comment, which we acknowledge is fundamental to improving the manuscript. We have thoroughly revised the paper to address the lack of a cohesive narrative and to better integrate the sub-studies.

And the study does not offer significant methodological innovation or new scientific insights and lacks discussion part. The only potentially unique aspect is the absence of previous atmospheric column observations in the suburban area of Xining city. However, this alone does not justify publication unless the authors can thoroughly address the concerns outlined above through major revisions.

We appreciate the referee's feedback and the concerns raised regarding the originality and scientific contribution of the study. We believe that our study presents a novel approach for estimating CO2 emissions by combining collaborative ground-based and space-based observations. While TROPOMI offers high spatial coverage for CO measurements, TROPOMI does not provide CO2 data, and satellites like OCO-2/3 or GOSAT have limited coverage in this region. The use of the EM27/SUN instrument, which probes both columnar CO and CO2 concentrations in the study area, is a key strength, as it allows for a more direct linkage between CO and CO2 emissions. Furthermore, the ground-based measurements offer a valuable dataset for validating satellite observations and improving model accuracy especially for this specific region, where satellite coverage is limited and which is difficult from the viewpoint of satellite measurements, as it is orographically complex.

We have emphasized this unique advantage and the potential of the proposed methodology to improve our understanding of urban emission sources in the revised manuscript. Furthermore, we have expanded the discussion section to address the broader scientific implications and how this combined approach could contribute to more accurate carbon cycle studies in regions with limited satellite data coverage.

**Major Comments:**

1. The observational period was very short, only 8 days in early June 2024, but the rationale for selecting this specific timeframe is unclear. Was there a particular emission event or atmospheric condition of interest during this period? The motivation for the campaign is not well explained. Additionally, the measurements were conducted in a suburban area, but it is not clear whether this site is representative, whether the data can inform future carbon cycle studies, or what the broader scientific significance is. Summer conditions are typically associated with various interfering factors, yet these are neither acknowledged nor discussed in the manuscript.

Thank you for the valuable comment. The choice of early summer for the observational period was intentional, as it helps minimize the influence of heating emissions from the cold season, which typically lead to higher CO emissions due to residential and industrial combustion. These emissions are difficult to separate from those of transportation and other sources during colder months. By selecting the early summer period, we aim to focus on emissions that are more representative of typical urban activity, without the confounding influence of winter heating.

Additionally, the EM27/SUN measurement site was located to the west of the city to capture emissions across the entire urban area, as the predominant wind direction in this region is easterly. This placement allows for a more comprehensive representation of the city's emission profile.

2. The manuscript aims to evaluate satellite retrievals using ground-based observations. In Section 2.2, a detailed description of the general COCCON product is provided, but there is almost no information about the specific EM27/SUN instrument used in this campaign. Key details such as the instrument's stability before and after the measurement period, the configuration of retrieval parameters, and whether any calibration was performed using TCCON, AirCore, or aircraft measurements are missing. Additionally, the measurement uncertainty is not discussed. As the ground-based observations serve as the reference for satellite validation, it is essential to present their accuracy and reliability clearly, rather than focusing only on general background information.

We thank the reviewer for highlighting the need for a more detailed description of the performance and quality assurance of EM27/SUN. In response, we have substantially expanded Section 2.2 of the manuscript to provide a comprehensive account of the specific EM27/SUN spectrometer used in this campaign, directly addressing the points raised concerning instrumental stability, retrieval configuration, calibration, and measurement uncertainty.

The EM27/SUN spectrometer used in this study operates within the COCCON network, whose instruments are well-established as robust and reliable through numerous international field campaigns (Butz et al., 2017; Frey et al., 2019; Klappenbach et al., 2015; Sha et al., 2025; Tu et al., 2020). Following the standard COCCON protocol, our specific instrument was calibrated prior to the campaign against the TCCON reference spectrometer at KIT. This calibration ensures accuracy by establishing and applying instrument-specific calibration factors for each target gas species (Frey et al., 2019).

To continuously monitor the instrument's stability and characterize its performance—key to assessing measurement uncertainty—we employed the two primary quality assurance methods proposed by the COCCON community: the Instrumental Line Shape (ILS) and the XAIR ratio. The ILS, which characterizes the spectrometer's spectral response function, is determined through laboratory open-path measurements using a nonlinear least-squares spectral fitting algorithm (Frey

et al., 2015; Hase et al., 1999). The XAIR ratio, an indicator of overall instrumental stability, is derived from the measured vertical columns of O2 and H2O in conjunction with surface pressure data (Alberti et al., 2022). Systematic monitoring of these parameters is critical for detecting any deviations from the expected instrumental performance.

Specifically, an ILS characterization was performed in December 2024. The results indicated a minor change: one key parameter, the modulation efficiency (ME), had decreased by approximately 1.5% compared to its baseline value measured at KIT. This deviation falls well within the accepted uncertainty range for such instruments. We attribute this small change primarily to mechanical stresses incurred during the long-distance shipment from Germany to China and subsequent domestic transports. A potential minor contribution from systematic errors associated with the specific light source and lens used in the ILS setup in China cannot be entirely ruled out. A follow-up ILS measurement in September 2025 showed no change in the ME value, confirming the instrument's stability.

Additionally, the XAIR ratio served as an independent indicator of instrumental performance throughout the measurement period. The mean XAIR value during the entire field campaign was  $1.0012 \pm 0.0024$ . Values consistently close to 1.0 signify highly stable instrument operation. This stability is further underscored by the nearly identical XAIR values measured before (1.0005) and after (1.0005) the campaign, indicating no significant instrumental drift.

In summary, we believe that the continuous monitoring of both ILS and XAIR provides a robust framework for quality assurance, effectively verifying the stability and reliability of the EM27/SUN spectrometer during the campaign. This approach is particularly valuable for ensuring data quality in remote field deployments where frequent re-calibration against a TCCON station is not feasible. All detailed information regarding the instrument's calibration, stability, retrieval parameters, and the associated quality control procedures has been incorporated into Section 2.2 as suggested.

3. The manuscript also attempts to evaluate CAMS simulation results using ground-based observations. However, the approach raises several questions. A 20 km radius was used for CAMS product validation—but why? Since CAMS provides data at specific grid points, the rationale for selecting a 20 km averaging radius is unclear. Large-area averaging is typically applied in satellite validation to reduce observational noise and improve sampling statistics, but it is not obvious why a 20 km radius was appropriate or necessary in this case.

We thank the referee for this critical point. The initial 20 km radius was indeed arbitrary. To address this, we have revised the collocation criteria to use the CAMS data from the nearest grid cells in which the EM27/SUN instrument was located. As the instrument was situated near the border of two grid cells, we used the average value from both. The relevant figures and text have been updated throughout the manuscript.

Moreover, the analysis in this section is not sufficiently developed. The authors conclude that CAMS performs well in simulating CH4 in this region, while its performance for CO and CO2 is poor. But what are the broader implications of this result? Does this indicate that CAMS is better suited for CH4 studies over the Qinghai-Tibetan Plateau? Could this support the case for establishing long-term observation sites in the area?

We thank the referee for this valuable suggestion. We have expanded the discussion in this section in the revised manuscript to address these findings:

"When comprising the absolute column amounts, CAMS shows an approximately 0.4% underestimation for  $XCO_2$  and 0.2% for  $XCH_4$  relative to the COCCON data. However, the underestimate of CO is more pronounced, with a bias of 35%.

CAMS XCH4 compares much better with COCCON observations than XCO2 and CO, as the higher value of  $R^2$  and the better agreement of the slope of the regression line indicates. We conclude that the variability of XCO and XCO2 dominating the variability as detected by the ground-based observation is generated on a smaller spatial scale which is not properly resolved in the simulation. In contrast, the variability of the CH4 model field seems to be dominated by extended sources distributed in a wider area, which therefore can be properly depicted by the model.

In terms of broader implications, the relatively strong performance of CAMS in simulating CH4, especially when compared to CO and CO2, indicates that CAMS may be more reliable for CH4 studies in such regions. This finding could potentially support the case for establishing long-term observation sites in this area to help with satellite validation and improve model accuracy, particularly for species like CO, which appear to require more refined simulations."

In addition, satellite retrievals usually offer a higher number of soundings than ground-based instruments. How does the CAMS product compare with satellite observations in terms of coverage and consistency? Do the conclusions drawn from CAMS agree with those from the COCCON dataset? These questions are not clearly addressed in the manuscript.

We appreciate the referee's valuable comment regarding the comparison between CAMS and satellite observations. To address this, we regridded TROPOMI data to the same spatial resolution as the CAMS GHG forecast over the study area. The figures below shows the correlation between these two datasets.

In general, CAMS agrees well with TROPOMI CO data, with CAMS showing approximately 3% higher CO amounts compared to TROPOMI. However, larger biases are evident over the northern and eastern parts of the study area, which correspond to the main valley region (see Figure 1-right below and Figure A5 in the manuscript). These discrepancies may be due to overestimation in the CAMS simulations or potential underestimation in the TROPOMI observations in the complex terrain of the region. This also highlights the importance of establishing long-term ground-based observations in such regions to better constrain model outputs and support satellite validation.

Figure 1 left: correlation between CAMS and TROPOMI for CO, right: spatial distribution of difference in columnar CO amounts between CAMS and TROPOMI.

For collocated XCH4, CAMS shows good agreement with TROPOMI, with a bias of -0.18%. However, the availability of TROPOMI XCH4 data in this region is limited (see Figure 2 right), with only about 3000 observations collected over five years. This relatively small dataset also makes it challenging to estimate CH4 emissions accurately from the TROPOMI dataset in this region.

Figure 2 left: similar to Figure 1-left, but for XCH4, right: total number of measurements in each grid during May 2018 – May 2024.

4. The manuscript estimates CO2 emissions using CO fluxes derived from TROPOMI and EM27/SUN observations. However, several points need clarification. First, the emission estimation appears to be based on multiplying values observed by wind speed. If so, does this method account for transport processes and particle dilution along the plume? A brief explanation of this approach in the Methods section would be helpful. In addition, how does the CO:CO2 ratio used in this study compare with values reported in emission inventories? This could be further discussed, especially in relation to combustion efficiency and source attribution.

Moreover, there appears to be an inconsistency in logic. In the CAMS evaluation, CO and CO2 simulations were shown to perform poorly, while CH4 agreed well with observations. However, in this section, the calculated CO emissions match CAMS values, and CO2 emissions align closely with inventory estimates. Given the earlier performance issues with CO and CO2 in CAMS, this raises questions about the reliability of the derived fluxes. What about CH4 emissions in this context? Without addressing this discrepancy, the conclusions are difficult to reconcile.

Thank the referee for these comments.

(1) The method considers the transport along the plume. We have added the explanation of this approach in Section 2.6:

**"2.6 Dispersion model and wind-assigned anomaly method**

For a single point source, the total emission is calculated by multiplying the measured total column enhancement ( $\Delta CO$ ) by the area of the affected plume (Babenhauserheide et al., 2020). This plume area is modeled as an evenly distributed cone, representing the long-term averaged dispersion (Tu et al., 2022a). The relationship is given by the following equation:

where  $\triangle CO$  represents the enhanced CO column observed at the downwind site, d is the distance from the source to the measuring site and v is the wind speed.

To estimated averaged emissions from satellite observations over a region, the wind-assigned method was applied (Tu et al., 2022a, 2022b, 2023, 2024b). This technique fits the anomalies between the satellite observations and the dispersion model by analyzing enhancements under opposing wind sectors. Specifically, the wind-assigned anomaly is defined as the difference in observed enhancements between two opposite wind fields (e.g., E: 0°–180° and W: 180°–360°). A key advantage of this approach is that it inherently eliminates the uncertainty associated with background concentration calculations for long-lived gases like CO, thereby significantly improving the reliability of the resulting emission estimates."

(2) A linear fit to the EM27/SUN observations yields a  $\Delta XCO/\Delta XCO_2$  slope of 0.035 ppb/ppb. This observed enhancement ratio is compared to the CAMS emission inventory ratio of 0.021 for CO/CO2 over the same region. In this study, we find that the CAMS model underestimates the forecast CO column by an average factor of 1.6127 compared to the ground-based observations. Compensating for this bias by scaling the CAMS CO emission rate increases the emission ratio to 0.034, which aligns closely with the observed enhancement ratio (0.035 ppb/ppb). We have added this ratio comparison to the manuscript.

Additionally, we have incorporated the following explanation regarding combustion into the manuscript in section 3.2:

"CO emissions from vehicle exhaust, a major contributor to air pollution, is closely related to fossil fuel combustion (Naus et al., 2018). Gao et al. (2025) reported that CO emissions increased significantly with altitude, observing nearly twice the emission levels in Xining (2320 m) compared to those at an altitude of 20 m. This trend can be attributed to the decline in atmospheric pressure and air density at higher elevations (Fattah et al., 2019). Under such conditions, engines draw in less air per cycle, which alters the air-fuel ratio and leads to suboptimal combustion. As altitude rises, the excess air ratio decreases because more diesel fuel is injected into the cylinders, but less air is captured per cycle. Therefore, the combustion between fresh air and diesel fuel becomes incomplete, resulting in substantial CO emissions."

(3) We thank the referee for raising this important point regarding the discrepancy between CO and CO2 emissions in the CAMS inventory and the column-based observations.

When directly comparing columnar gas ratio between COCCON and CAMS, we find that the ratios for XCO2, XCH4, and CO are 1.0042, 1.0022, and 1.6127, respectively. As discussed in (2), adjusting the CAMS CO emission rate by a factor of 1.6127 compensates for the discrepancy in CO, resulting in an emission ratio between CO and CO2 that aligns more closely with the column ratio observed by the EM27/SUN measurements.

It is also important to note that the calculated CO emission rate is based on TROPOMI XCO data, which is approximately 33% underestimated compared to the EM27/SUN measurements. The average ratio between EM27/SUN and TROPOMI is 1.54, similar to that between EM27/SUN and CAMS. Adjusting for this underestimation also brings the CO/CO2 emission ratio more in line with the column ratio observed by EM27/SUN.

As discussed in Comment #3, the limited TROPOMI XCH4 data in this region makes it challenging to estimate emission rates accurately. Furthermore, the correlations between XCH4 and XCO2 or XCO from the ground-based observations are relatively weak (see Figure 3), complicating the estimation of CH4 emissions based on these correlations. Longer observation periods may help improve these correlations and provide more reliable CH4 emission estimates.

Figure 3 correlation between ΔXCO and ΔXCH4 (left), and between ΔXCH4 and ΔXCO (right).

**Minor Comments:**

Line 53–55: I still do not fully understand the claim that surface observations are influenced by surface exchange but limit the ability to estimate sources and sinks. Given that surface measurements are sensitive to near-surface fluxes, wouldn't they actually be more effective for detecting local sources and sinks? This statement needs clarification.

The referee is correct in noting that surface observations are more sensitive to near-surface fluxes and are therefore effective in representing local sources and sinks, particularly at smaller spatial scale. In contrast, column-based measurements, such as those from FTIR observations, are better suited for capturing source and sink information over broader, regional areas. We acknowledge that the original wording may have been misleading, and have revised the sentences accordingly:

"These stations conduct in-situ measurements and provide highly accurate surface observations and valuable insights into local fluxes; however, they are influenced by surface exchanges and vertical transport, which can limit their ability to estimate sources and sinks over larger spatial scales. When combined with other observation types, such as FTIR, which capture emissions and transport on a broader scale, they become complementary, together offering a more comprehensive understanding of sources and sinks at both local and regional levels (Callewaert et al., 2022)."

Line 161: This point raises concerns. The measurement period spans only 8 days, while coal mining activity is often highly episodic. As shown in the study by [Author] (https://doi.org/10.1016/j.ecolind.2023.110454), regions similar to the study area in Qinghai are known to have significant coal mining emissions. Therefore, concluding that there is no such influence based solely on this short observational window appears premature.

We thank the referee for raising this important point. As suggested, we have removed the premature conclusion from line 161 and have added a discussion on this limitation and the cited the abovementioned literature in the revised manuscript.

"The average observed XCH4 concentration during the measurement period was  $1898.45 \pm 6.66$  ppb. Notably, 300 km to the northwest of the study site lies the largest Muli coalfield in Qinghai Province, which has an estimated coal reserve of around 4 billion tons (Xiao et al., 2023). However, the maximum XCH4 enhancement observed was approximately 10 ppb on May 25, when the wind was from northwest before noon. This enhancement is relatively lower than those reported in other coal fields, such as in Changzhi, Shanxi Province (Tu et al., 2024a), suggesting that the ground-based observations may not have fully captured the methane emissions. The modest enhancement is likely due to the relatively greater distance from the source, combined with potentially low coal mining activity during the observing period, as coal production is often highly episodic."

Butz, A., Dinger, A. S., Bobrowski, N., Kostinek, J., Fieber, L., Fischerkeller, C., Giuffrida, G. B., Hase, F., Klappenbach, F., Kuhn, J., Lübcke, P., Tirpitz, L., and Tu, Q.: Remote sensing of volcanic CO2, HF, HCl, SO2, and BrO in the downwind plume of Mt. Etna, Atmos. Meas. Tech., 10, 1–14, https://doi.org/10.5194/amt-10-1-2017, 2017.

Frey, M., Sha, M. K., Hase, F., Kiel, M., Blumenstock, T., Harig, R., Surawicz, G., Deutscher, N. M., Shiomi, K., Franklin, J. E., Bösch, H., Chen, J., Grutter, M., Ohyama, H., Sun, Y., Butz, A., Mengistu Tsidu, G., Ene, D., Wunch, D., Cao, Z., Garcia, O., Ramonet, M., Vogel, F., and Orphal, J.: Building the COllaborative Carbon Column Observing Network (COCCON): long-term stability and ensemble performance of the EM27/SUN Fourier transform spectrometer, Atmos. Meas. Tech., 12, 1513–1530, https://doi.org/10.5194/amt-12-1513-2019, 2019.

Klappenbach, F., Bertleff, M., Kostinek, J., Hase, F., Blumenstock, T., Agusti-Panareda, A., Razinger, M., and Butz, A.: Accurate mobile remote sensing of XCO2 and XCH4 latitudinal transects from aboard a research vessel, Atmos. Meas. Tech., 8, 5023–5038, https://doi.org/10.5194/amt-8-5023-2015, 2015.

Sha, M. K., Das, S., Frey, M. M., Dubravica, D., Alberti, C., Baier, B. C., Balis, D., Bezanilla, A., Blumenstock, T., Boesch, H., Cai, Z., Chen, J., Dandocsi, A., Mazière, M. D., Foka, S., García, O., Gillespie, L. D., Gribanov, K., Gross, J., Grutter, M., Handley, P., Hase, F., Heikkinen, P., Humpage, N., Jacobs, N., Jeong, S., Karppinen, T., Kiel, M., Kivi, R., Langerock, B., Laughner, J., Lopez, M., Makarova, M., Mermigkas, M., Morino, I., Mostafavipak, N., Nemuc, A., Newberger, T., Ohyama, H., Okello, W., Osterman, G., Park, H., Pirloaga, R., Pollard, D. F., Raffalski, U., Ramonet, M., Sepúlveda, E., Simpson, W. R., Stremme, W., Sweeney, C., Taquet, N., Topaloglou, C., Tu, Q., Warneke, T., Wunch, D., Zakharov, V., and Zhou, M.: Fiducial Reference Measurements for Greenhouse Gases (FRM4GHG): Validation of Satellite (Sentinel-5 Precursor, OCO-2, and GOSAT) Missions Using the Collaborative Carbon Column Observing Network (COCCON), Remote Sensing, 17, 734, https://doi.org/10.3390/rs17050734, 2025.

Tu, Q., Hase, F., Blumenstock, T., Kivi, R., Heikkinen, P., Sha, M. K., Raffalski, U., Landgraf, J., Lorente, A., Borsdorff, T., Chen, H., Dietrich, F., and Chen, J.: Intercomparison of atmospheric CO2 and CH4 abundances on regional scales in boreal areas using Copernicus Atmosphere Monitoring Service (CAMS) analysis, COllaborative Carbon Column Observing Network (COCCON) spectrometers, and Sentinel-5 Precursor satellite observations, Atmospheric Measurement Techniques, 13, 4751–4771, https://doi.org/10.5194/amt-13-4751-2020, 2020.

---

## Author Comment (AC2)

**Response to Referee #2**

We would like to thank reviewer #2 for taking the time to review this manuscript and for providing valuable, constructive feedback and corresponding suggestions that helped us to further improve the manuscript.

In this author's comment, all the points raised by the reviewer are copied here one by one and shown in black color, along with the corresponding reply from the authors in blue.

This paper by Tu et al., focus on observations at an industrial park and simulations of Xining's emissions using portable Fourier transform spectrometer and TROPOMI observations. The topic is interesting and falls into the scope of ACP. I have some major comments that may improve the quality of this paper.

**Major concerns:**

May-June may be too short to represent the whole year, and in winter there are coal-burning period for heating. Do the authors have longer time observations? Please at least add some discussions on this time coverage influences.

We thank the referee for raising this important point. Our study is indeed based on a three-week intensive campaign, and the number of valid observation days was further reduced due to unfavorable weather conditions.

We acknowledge that the limited time coverage may not fully capture the seasonal variability, especially during the winter heating period. We have now included a discussion in the revised manuscript addressing this limitation and its possible influence on the representativeness of our results:

"The observed discrepancies compared with inventories may be attributed to differences in temporal coverage, methodological approaches, and potential changes in emission patterns over time. Additionally, it should be noted that the field campaign spanned only three weeks from May to June, which mainly represents early summer. During other seasons, such as summer or winter, when photosynthesis activities or coal burning for heating is more prevalent, the  $\Delta XCO/\Delta XCO_2$  ratios and associated  $CO_2$  emissions may differ. A longer period of ground-based observations and running several spectrometers upwind and downwind may improve our results. Our findings so far demonstrate the potential of the EM27/SUN spectrometer as a promising tool for comprehensively evaluating greenhouse gas (GHG) and air pollutant emissions in urban areas (Che et al., 2022b; Lee et al., 2024)."

CAMS resolution and emissions information may be too sparse to include local emission areas and may not be appropriate for the comparison.

CAMS inventory has a relatively high spatial resolution of  $0.1^{\circ} \times 0.1^{\circ}$ , which allows for reasonably fine-scale emission estimates. Our comparison is based on TROPOMI CO data over a regional area, making the datasets generally comparable.

As also noted by another referee, we have revised the collocation criteria in the updated manuscript to use the CAMS data from the nearest grid cells to the location of the EM27/SUN instrument. The relevant figures and text have been updated accordingly.

I suggest the authors include analyses and comparisons with open accessed inventory (e.g. MEIC). And add some discussions on the difference between inventory and inversions.

We have added discussion about MEIC inventory to section 3.5. A figure presenting emission from this study and different inventories has been added to the manuscript:

Figure 5: CO and CO2 emissions from this study and different inventories. The red start symbols represent the highest value derived from EM27/SUN observations.

**More discussions are also added in the manuscript:**

"We estimate an average  $CO_2$  emission rate of approximately 550 kg/s, which aligns well with the CAMS-GLOB-ANT (617 kg/s for 2020) though lower than the Carbon Emission Accounts and Datasets (CEADs) (726 kg/s for 2015) ("Methodology and applications of city level  $_{CO2}$  emission accounts in China," 2017; Shan et al., 2018) and MEIC (935 kg/s for 2020) estimates. The data also reveal strong daily fluctuations in emissions. The peak event was observed on May 27, which exhibited a maximum  $\Delta XCO:\Delta XCO_2$  ratio of 40.08 ( $R^2=0.8544$ ). This ratio translates to a maximum  $CO_2$  emission rate of 55.6 kg/s and a concurrent maximum  $CO_2$  emission rate of 2180 kg/s.

Additionally, the CAMS and MEIC inventories show similar  $CO/CO_2$  emission ratios of 0.021 and 0.018, respectively. As detailed in Section 3.3, both TROPOMI and CAMS underestimates the atmospheric CO column by a factor of approximately 1.6. When we correct for this bias by scaling the TROPOMI-derived emission and CAMS inventory, the resulting emission ratio increases to 0.034. This corrected value aligns closely with our ground-based observed  $\Delta XCO/\Delta XCO_2$  enhancement ratio of 0.035 ppb/ppb."

Spatial distributions associated with the TROPOMI data, simulations and inversions are needed to improve the content of this paper.

Thank the referee for this suggestion. In section 3.4, we have addressed this by applying a multiyear inversion of emissions based on TROPOMI data, using a dispersion model coupled with a wind-assigned anomaly method.

Our method uses a dispersion model driven by wind fields and a priori emissions to simulate plume enhancements. A wind-assigned anomaly technique is then applied to both the TROPOMI data and the model simulations. This technique calculates the difference in enhancements under opposing wind conditions, effectively removing background bias. The final emission inversion is derived by scaling the a priori emissions to minimize the difference between the modeled and observed anomalies. We have added the explanation of this approach in Section 2.6:

**"2.6 Dispersion model and wind-assigned anomaly method**

For a single point source, the total emission is calculated by multiplying the measured total column enhancement ( $\Delta CO$ ) by the area of the affected plume (Babenhauserheide et al., 2020). This plume area is modeled as an evenly distributed cone, representing the long-term averaged dispersion (Tu et al., 2022a). The relationship is given by the following equation:

$$\varepsilon = \Delta CO \times d \times v \times \partial$$
 Eq.

where  $\Delta CO$  represents the enhanced CO column observed at the downwind site, d is the distance from the source to the measuring site and v is the wind speed.

To estimated averaged emissions from satellite observations over a region, the wind-assigned method was applied (Tu et al., 2022a, 2022b, 2023, 2024b). This technique fits the anomalies between the satellite observations and the dispersion model by analyzing enhancements under opposing wind sectors. Specifically, the wind-assigned anomaly is defined as the difference in observed enhancements between two opposite wind fields (e.g., E: 0°–180° and W: 180°–360°). A key advantage of this approach is that it inherently eliminates the uncertainty associated with background concentration calculations for long-lived gases like CO, thereby significantly improving the reliability of the resulting emission estimates."

**Besides the CO and CO2 emissions rates, the CH4 emissions rates are also important.**

We appreciate the referee's comment and agree that CH4 emissions are indeed important. Ground-based FTIR measurements did capture CH4 concentrations. However, we found that the  $\Delta$ XCH4 does not exhibit a consistent correlation with  $\Delta$ XCO or  $\Delta$ XCO2 (Figure 1), unlike the more stable relationship between  $\Delta$ XCO and  $\Delta$ XCO2. This consistent correlation between CO and CO2 suggests co-emission, aiding the reliability of CO2 emission estimates from CO. In contrast, the variable correlation for CH4 introduces greater uncertainty in estimating its emissions using the same methodology applied to CO. Longer observation periods may help improve these correlations and refine CH4 emission estimates. Additionally, the weaker correlation between  $\Delta$ XCH4 and the other species may indicate that CH4 is not significantly co-emitted with CO and CO2.

Figure 1: correlation between ΔXCO and ΔXCH4 (left), and between ΔXCH4 and ΔXCO (right).

Additionally, the availability of TROPOMI XCH4 data in this region is limited (see Figure 2), with only about 3000 observations collected over five years. This relatively small dataset also makes it challenging to estimate CH4 emissions accurately from the TROPOMI dataset in this region.

Figure 2 total number of measurements in each grid during May 2018 - May 2024.

It is also important to note that the primary focus of this study is on connecting satellite and ground-based remote sensing observations, specifically by estimating  $CO_2$  emissions from the ground-based observed  $\Delta XCO/\Delta XCO_2$  ratio particularly when  $CO_2$  observations are sparse. As such,  $CH_4$  emissions are not addressed in this analysis.

**Minor comments:**

Add serial numbers to the subFigures in Fig.1, and the font in subFigure2 is too small and difficult to read.

Thanks. Figure 1 has been updated.

line159: Does this sentence means that CO and CO2 come from different sources?

For better clarification, we have revised the sentence to:

"The enhancement of XCO and XCO2 ratio ( $\Delta$ XCO2, see section 3.5) exhibited slopes of 14.43 ppb/ppm before noon and 4.76 ppb/ppm in the late afternoon. Both values were significantly lower than those observed under easterly wind conditions. This suggests that the CO and CO2 emissions in the western regions originate from different combustion processes or source types compared to those in the east."

Add more descriptions for Fig.2 (a,b,c). And for Figure3, do data from TROPOMI (5.5 km × 7 km) and COCCON (point) have comparable spatial representativeness? What processing methods were applied? These should be explicitly stated in the Methods and in discussions.

We thank the referee for this suggestion. We have expanded the discussion on Fig2 in section 3.1.

COCCON is a network of ground-based remote sensing FTIR spectrometers that supplements the existing TCCON stations. Like TCCON, COCCON provides column-averaged concentrations that are directly representative of the local conditions above the measurement site. To ensure a robust comparison between the point measurements from COCCON and the integrated area measurements from TROPOMI, it is crucial to apply appropriate spatial and temporal collocation criterion. This method is well-established in the literature for satellite validation (e.g., Klappenbach et al 2015, Velazco et al 2019, Tu et al 2020, Knapp et al 2021, Alberti et al 2022, Sha et al 2024).

Various studies have applied different spatiotemporal criteria based on the characteristics of the satellite data and ground-based measurements. For example, Klappenbach et al. (2015) used a 5° latitudinal/longitudinal radius and a 4-hour temporal window for GOSAT overpasses. More stringent criteria have been used in subsequent studies, including a  $100 \, \text{km} - 200 \, \text{km}$  spatial rediuas and a  $\pm 1$  to  $\pm 2$  hour temporal window for GOSAT (Velazco et al 2019) and TROPOMI (Tu et al 2020, Sha et al 2025).

In this study, to ensure sufficient data pairs for robust validation, we applied the following collocation criteria:

- o Spatial: a 200 km radius for XCH4 and 100 km radius for XCO.
- Temporal: a ±2h window around the COCCON measurements to align with TROPOMI overpasses.

These criteria are stated in section 3.2 of the manuscript.

lines 193-195: Why not match the COCCON data with the grid scale of CAMS? At distances beyond 20km or even 50km, and the factors influencing observations or forecast results are local emission sources and atmospheric transport processes.

We thank the referee for this comment. The CAMS forecast data have a spatial resolution of  $0.1^{\circ}\times0.1^{\circ}$ . We have revised the collocation criteria to use the CAMS data from the nearest grid cells in which the EM27/SUN instrument was located. The relevant figures and text have been updated throughout the manuscript.

Figure 4b: The data points are overly clustered. It is recommended to reduce the range of the x-y axes, for instance to 1870-1950. And other subplots also need to be improved for this aspect.

Thanks. We have updated this figure.

Figure4c: The legend should not overlay the data plots.

Thanks. We have updated this figure.

Line 205: To what extent is this underestimation a result of observation? Have you considered spatial representativeness inconsistency as a potential source?

This underestimation in satellite observations might due to errors at higher altitude

Line 217: Enhanced relative to what?

The enhanced CO column is relative to the background, i.e., representing the emitted CO.

lines 225-227: The definitions of background CO concentration and  $\Delta$ XCO should be provided when these terms were firstly appeared.

Thank the referee. We have provided the definitions when these terms were firstly appeared in the manuscript.

Line 241-242: Has the higher emissions led to the observed concentration peak?

The referee is right that the higher emissions contribute to observed peak on this short time ( $\sim$ 1h). To observe concentration peaks is also largely due to the wind direction. Peaks are easily observed when obverse site is exactly in the downwind of the sources and the wind is steady.

12 Why only analyze the CO emission and the relation of  $\Delta$ XCO and  $\Delta$ XCO2? How about CH4?

Thank the referee for raising this point. The XCO and XCO2 enhancements show a clear correlation, reflecting their co-emission from fossil fuel combustion and biomass burning, especially in urban region. However, we did not find a consistent correlation between CH4 and either CO2 or CO (as discussed in the major concerns), suggesting that CH4 emissions in the study region are influenced by additional sources beyond combustion.

13 Please have the manuscript polished again for grammar and spellings.

Thanks. We have tried our best to modify the manuscript.

- Alberti C, Tu Q, Hase F, Makarova M V, Gribanov K, Foka S C, Zakharov V, Blumenstock T, Buchwitz M, Diekmann C, Ertl B, Frey M M, Imhasin H Kh, Ionov D V, Khosrawi F, Osipov S I, Reuter M, Schneider M and Warneke T 2022 Investigation of spaceborne trace gas products over St Petersburg and Yekaterinburg, Russia, by using COllaborative Column Carbon Observing Network (COCCON) observations *Atmos. Meas. Tech.* **15** 2199–229
- Klappenbach F, Bertleff M, Kostinek J, Hase F, Blumenstock T, Agusti-Panareda A, Razinger M and Butz A 2015 Accurate mobile remote sensing of XCO2 and XCH4 latitudinal transects from aboard a research vessel *Atmos. Meas. Tech.* **8** 5023–38
- Knapp M, Kleinschek R, Hase F, Agustí-Panareda A, Inness A, Barré J, Landgraf J, Borsdorff T, Kinne S and Butz A 2021 Shipborne measurements of XCO2, XCH4, and XCO above the Pacific Ocean and comparison to CAMS atmospheric analyses and S5P/TROPOMI *Earth Syst. Sci. Data* 13 199–211
- Sha M K, Das S, Frey M M, Dubravica D, Alberti C, Baier B C, Balis D, Bezanilla A, Blumenstock T, Boesch H, Cai Z, Chen J, Dandocsi A, Mazière M D, Foka S, García O, Gillespie L D, Gribanov K, Gross J, Grutter M, Handley P, Hase F, Heikkinen P, Humpage N, Jacobs N, Jeong S, Karppinen T, Kiel M, Kivi R, Langerock B, Laughner J, Lopez M, Makarova M, Mermigkas M, Morino I, Mostafavipak N, Nemuc A, Newberger T, Ohyama H, Okello W, Osterman G, Park H, Pirloaga R, Pollard D F, Raffalski U, Ramonet M, Sepúlveda E, Simpson W R, Stremme W, Sweeney C, Taquet N, Topaloglou C, Tu Q, Warneke T, Wunch D, Zakharov V and Zhou M 2025 Fiducial Reference Measurements for Greenhouse Gases (FRM4GHG): Validation of Satellite (Sentinel-5 Precursor, OCO-2, and GOSAT) Missions Using the COllaborative Carbon Column Observing Network (COCCON) *Remote Sensing* 17 734
- Sha M K, De Mazière M, Notholt J, Blumenstock T, Bogaert P, Cardoen P, Chen H, Desmet F, García O, Griffith D W T, Hase F, Heikkinen P, Herkommer B, Hermans C, Jones N, Kivi R, Kumps N, Langerock B, Macleod N A, Makkor J, Markert W, Petri C, Tu Q, Vigouroux C, Weidmann D and Zhou M 2024 Fiducial Reference Measurement for Greenhouse Gases (FRM4GHG) *Remote Sensing* 16 3525
- Tu Q, Hase F, Blumenstock T, Kivi R, Heikkinen P, Sha M K, Raffalski U, Landgraf J, Lorente A, Borsdorff T, Chen H, Dietrich F and Chen J 2020 Intercomparison of atmospheric CO2 and CH4 abundances on regional scales in boreal areas using Copernicus Atmosphere Monitoring Service (CAMS) analysis, COllaborative Carbon Column Observing Network (COCCON) spectrometers, and Sentinel-5 Precursor satellite observations *Atmospheric Measurement Techniques* 13 4751–71
- Velazco V A, Deutscher N M, Morino I, Uchino O, Bukosa B, Ajiro M, Kamei A, Jones N B, Paton-Walsh C and Griffith D W T 2019 Satellite and ground-based measurements of XCO2 in a remote semiarid region of Australia *Earth Syst. Sci. Data* 11 935–46